# The Double Mediating Effect of Family Support and Family Relationship Satisfaction on Self-Compassion and Meaning in Life among Korean Baby Boomers

**DOI:** 10.3390/ijerph19169806

**Published:** 2022-08-09

**Authors:** Yu-soo Jeong, Young-soon Lee

**Affiliations:** Department of Psychology, Jeonbuk National University, Jeonju-si 54896, Korea

**Keywords:** Korean baby boomer, self-compassion, meaning in life, family support, family relationship satisfaction

## Abstract

This study identified the relationship between self-compassion and meaning in life among Korean baby boomers and examined the double mediating effect of family support and family relationship satisfaction on this relationship. For this purpose, data were collected from 400 baby boomers (born between 1955–1963) using the self-compassion, meaning in life, family support, and family relationship satisfaction scales. PROCESS Macro 3.5 Model 6 was used to analyze the double mediating effects. The results revealed that first, there was a significant correlation between the self-compassion, meaning in life, family support, and family relationship satisfaction of this study. Second, in the relationship between self-compassion and the meaning in life, family support, and family relationship satisfaction were found to have a partial mediating effect and a double mediating effect. The implications and limitations of these findings are also discussed.

## 1. Introduction

Korean baby boomers refers to the large group of people who were born between 1955 when the fertility rate rose sharply after the Korean War and 1963 when the birth rate slowed due to birth control policies, accounting for about 14.5% of the population [1]. Those who began to enter the aged society in 2020 tend to have a high level of education and professionalism, accept old age as an opportunity for self-realization, and have high participation in social activities [2]. In addition, compared to the older generation, they have better access to information, the ability to find information on their own, and the ability to utilize various types of information [3].

As such, the baby boomers have a distinct strength from the previous generation, but they are called the ‘Sandwich generation’ and are negatively affected by the economic income cliff [4] due to the dual stimulus burden of parental care and child rearing. Because they contributed significantly to productivity compared to the previous generation [5], they experience a significant decline in their sense of value, such as loss of presence in a situation where they can no longer make productive contributions as they enter old age.

In the past, work in Korea was considered a means of living or a duty. However, in modern times, the meaning of work has expanded to have a close relationship with overall life satisfaction and meaning, psychological prosperity, mental health, and happiness. In addition, today’s work is a person’s identity and a factor that predicts the future, and the proportion of human life is increasing unprecedentedly. A recent study revealed that the meaning of work in Korea has expanded beyond the meaning of economic means and family support to life vitality, growth opportunities, interpersonal relationships, and social contributions [6]. Now, Korea has entered the age of 100 years of life expectancy and is about to enter a super-aged society. Therefore, it is a significant crisis that the baby boomers live a long old age after retiring from work, which affects life satisfaction, happiness, and mental health. Owing to this reason, they were found to have higher levels of stress compared to the previous generations [2].

Stress from negative emotional experiences in old age can act as a risk factor that can increase depression and lead to suicide [7]. In fact, in 2011, 10 years ago, the suicide rate of baby boomers per 100,000 people was 39.1 [8], but the suicide rate in 2019 rose significantly to 46.6 per 100,000 [9], which is much higher than the OECD (Organization for Economic Cooperation and Development) average of 17.2. This increase in the suicide rate suggests that there is an urgent need for intervention measures in terms of mental health for older adults in Korea, including baby boomers.

According to the Interpersonal Psychological Theory of Suicide (IPTS), frustration with belonging and the perceived burden on others are two important factors that make up the desire to commit suicide [10]. In fact, economic loss increased suicidal thoughts [11] and the suicide rate of older adults who were isolated or recently moved to live in the city was high [12]. It can be inferred that the high suicide rate among older adults, including baby boomers, is also caused by the increasing perceived burden, as retirement prevents them from making a productive contribution, and loneliness caused by a shrinking social network [13].

This study focused on meaning in life as an alternative to solving psychological difficulties that can drive older adults to suicide. Meaning in life is the process of discovering and pursuing meaning independently throughout life along with the integrated recognition of life in social relationships [14]. Older adults experience a decline in life satisfaction, due to a decrease in income that comes with retirement, as well as a declining physical state [15,16].

Older people experience loss of meaning in their lives and feel anxious about the rest of their lives [17], and the accumulation of these experiences increases the likelihood of their experiencing existential emptiness [18]. Individuals who experience lethargy and emptiness due to the loss of meaning in life may adopt a passive attitude or seek undesirable solutions without attempting to overcome negative situations [19]. In addition, drug and alcoholism, violence, impulsive behavior, other abnormal behaviors, or thinking of suicide easily in stressful situations [20,21] can lead to suicide in severe cases [22]. As such, the loss of meaning in life can cause maladjustment in many aspects, threatening mental health.

It is important to rediscover the value and meaning in life to overcome the existential void that endangers old age [23]. This is because satisfaction with the meaning in life affects positive attitudes and emotions about life, such as mental health, life satisfaction, generous attitudes toward others, and gratitude [24,25]. Old age is a time when meaning in life has a greater influence on health and happiness than in other generations, and it plays an important role in overcoming the difficulties and crises faced in old age [26,27]. According to the meta-analysis results related to the meaning among older adults, it was found that meaning in life had a stronger effect on positive factors than on negative factors [28]. These results show that meaning in life contributes to the improvement of psychological well-being by invoking the attention of older adults, who tend to focus on negative events experienced due to aging and helping them pay attention to psychological resources that are already acquired.

It is important to discover and pursue meaning in life in old age, but in old age, it is not easy to rediscover meaning in life due to many negative experiences, such as the deterioration of health caused by physical and physiological functions, the reduction of income because of retirement, the loss of social role, and the loss of ties. In fact, many older adults are known to experience loneliness, alienation, a decrease in self-esteem, and regret, thinking that there is no meaning in their lives anymore [29]. When negative evaluations, such as regret over the past are overwhelming, older adults are more likely to criticize themselves as their self-esteem decreases [30].

Self-criticism is a negative and harsh self-assessment process [31], in which repeated self-criticism makes an individual feel inadequate and progressively increases an adversarial and harsh manner toward oneself, which can worsen pathologically [32,33]. Self-criticism is a diagnostic factor that causes and maintains post-traumatic stress disorder [34] as well as social anxiety [35,36], suicide, and self-harm behavior [37,38]. In addition, because self-criticism causes depression [39] and affects suicide ideation [40,41], an alternative is needed to mediate this for healthy old age.

Self-compassion has been proposed as an intervention to prevent self-criticism. Self-compassion is a concept that approaches or treats oneself openly regarding pain, understands oneself when experiencing pain and failure, treats oneself kindly in an uncritical manner, and understands that individual experience is a universal aspect of life [42]. People with high levels of self-compassion were found to have high optimism, happiness, positive emotions, and subjective well-being. Self-compassion also has a positive effect on emotional intelligence, emotional regulation, social bonds, and social mastery [43,44]. In addition, it improves interpersonal skills [45,46] and contributes to improving the quality of marital relationships [47].

Several recent studies have also demonstrated that self-compassion can influence individuals to positively shape and recognize the meaning of their life [44,48]. Self-compassion interventions also lead to neuropsychological changes that effectively reduce immune system responses, negative affective responses, and stress hormones to psychosocial stress [49].

As such, it can be said that older adults with the idea of self-compassion are psychologically and mentally stable compared to other older adults and could lead a healthy life [50]. However, even though self-compassion has a positive effect on aging, domestic research so far has been conducted only on college students or general adults [51]. Therefore, this study attempted to identify parameters that can change the relationship between self-compassion and the meaning in life of baby boomers who began to enter old age and to suggest counseling interventions based on those parameters.

In this study, the relationship between the variables was examined by setting family support and family relationship satisfaction as parameters among the psychological mechanisms that can change in the process of rediscovering meaning in life for baby boomers who need self-compassion. This study attempts to analyze family relations, not social relations, for the following reasons. In old age, the support of informal networks is very important because the official social network is reduced for various reasons, such as retirement [13]. Among informal networks, family is the most important social support network in old age and plays a very important role in preserving the mental health of older adults [52].

In addition, family relationships are important because they become the social capital of family members, while providing the emotional support, material resources, and cultural values that family members need [53]. As described above, because one’s social network diminishes in old age, it is necessary to form high-quality relationships with one’s family. Furthermore, familial relationships provide the ability to smoothly support relationships other than family [54], so they should be dealt with carefully for limited relationship satisfaction. Therefore, family support and family relationship satisfaction need to be viewed as important resources when discussing the quality of life of old baby boomers.

The first parameter, family support, is defined as a social support system that lasts throughout life, adapting to crises when they occur, and plays a role in ensuring that family members in crisis have the power to overcome their situation [55]. Older family support plays a role in social support and has been found to reduce the harmful effects of stress and psychological and social burdens that directly affect health [56]. In addition, family support for older adults has a significant influence on depression [57]. However, it is difficult to find a study on the relationship between self-compassion and family support, which indicates that research on compassion is still in the nascent stage [58]. The relationship between the two can be conceptually inferred based on variables, such as self-compassion and family support, and studies [59,60,61] have revealed a positive correlation between self-compassion and other support (social and emotional support).

Family support has contributed to beneficial results in most adults in various areas [62]. It is associated with low morbidity and mortality [63], and high levels of family support have been found to increase happiness and lower the risk of depression [64]. It also allows individuals to cope effectively with and overcome personal burnout and stressful situations [65,66]. Several studies have shown that family support generally plays a key role in the functioning of individuals with chronic illness [67,68]. Thus, family support plays a crucial role in reducing or recovering from physical and psychological damage experienced by older adults.

According to previous studies, family support has a significant influence on the quality of life and the successful aging of older adults [69,70,71] and the level of meaning in life [72]. However, there are conflicting studies that show that family support does not significantly affect the life satisfaction of older adults [73]; therefore, it is necessary to check whether family support has a significant effect on the level of meaning in life. It is necessary to consider that baby boomers have two-sided values, of tradition and modernity, for their families. This is because they accept Western values through the socialization process and fulfill their responsibilities for parenting, but they do not want to burden their children when they become old [74]. Therefore, this study attempted to determine whether the family support for has a significant effect on meaning in life for baby boomers.

The second parameter, family relationship satisfaction, is a source of meaning in life [75], which refers to the overall level of satisfaction that family members feel during interactions in family life [76]. In a study involving older adults, family relationships ranked highest in terms of meaning in life [77], and older adults over 60 in Korea also ranked family-related areas (child relations, family bonds, marital status, and family support) as the most important areas of life [78]. Family relationships affect us throughout our lives [79], especially in old age, where these relationships remain relatively stable compared to neighbors and friends [80]. Therefore, older adults usually include their families in decision making when they encounter problems due to stress [81].

As old age is a time when dependence on family increases, and the perceived satisfaction of family relationships is very important. According to previous studies on family relationship satisfaction, self-compassion is positively correlated with family relationship satisfaction [82] and the level of family support affects family relationship satisfaction [83]. In addition, the better the family relationship, the lower the problem behaviors, and the higher the quality of life [84], causing physical function improvement and the emotional problems of older adults decrease [85,86]. In addition, it was found that social integration was easily achieved when family relationship satisfaction was high [87] and helped prevent emotional problems, such as depression [88]. Therefore, it is expected to have a positive effect on the improvement of meaning in life when the satisfaction level of family relationships perceived by the baby boomers, who have entered an aging society, is high.

Based on the above previous studies, it was assumed that there is a possibility that family support and family relationship satisfaction have a mediating effect on the relationship between self-compassion and meaning in life of baby boomers. Therefore, we hypothesized that there would be a double mediating effect of family support and family relationship satisfaction on the relationship between self-compassion and the meaning in life of baby boomers (Figure 1).

## 2. Methods

### 2.1. Participants

Data collection was commissioned by Invight Panel Co., Ltd. (Seoul, Korea), a panel data collection company, to respond to a survey of baby boomers (born between 1955–1963) living all over Korea. A survey was conducted with 400 participants, 200 men and women each, to examine the population by making the ratio of men to women the same. The survey was conducted online, and it collected data on demographic characteristics, self-compassion, meaning in life, family support, and family relationship satisfaction measures. The data from 400 questionnaires were used for the analysis.

#### Characteristics of Participants in This Study

The demographic and sociological characteristics of the participants and the levels of the major variables were as follows: Average age was 64.7 (*SD* = 4.54) of the 400 participants, 200 were male (50%) and 200 were female (50%); 385 (96.2%) were married, 15 (3.8%) were unmarried; and 227 (56.7%) were retired, 173 (43.3%) were not retired. Finally, by education level, 231 (57.8%) were college graduates, 105 (26.2%) were high school graduates or lower, and 64 (16.0%) were graduate school or higher.

### 2.2. Measures

#### 2.2.1. Self-Compassion

Self-compassion was measured using the Korean version of the Self-Compassion Scale (K-SCS) developed by Neff [89] and adapted and validated by Kim, Yi, Cho, Chai, and Lee [90] in Korean. The K-SCS consists of 26 items and two sub-factors: self-compassion (13 items, e.g., I treat myself with the care and tenderness I need when I go through really hard times.) and self-criticism (13 items, e.g., I tend to disapprove and criticize my own shortcomings and weaknesses.). Each item is rated on a five-point Likert scale ranging from 1 (not at all true) to 5 (very true). In this study, the internal consistency of the items (Cronbach’s α) was 0.88.

#### 2.2.2. Family Support

To measure family support, a scale developed by Kang, Seo, and Lee [91] for patients with partial paralysis was used, and one question that did not match the contents of this study was excluded. This scale consists of 10 items and two sub-factors: Positive part (8 items, e.g., My family cares and loves me.) and negative part (2 items, e.g., My family thinks I’m a nuisance.). Each item is rated on a five-point Likert scale ranging from 1 (not at all true) to 5 (very true). The internal consistency of the items (Cronbach’s α) was 0.94.

#### 2.2.3. Family Relationship Satisfaction

The participants’ level of family relationship satisfaction was measured using the Family Relationship Scale-15 developed by Yang and Kim [92]. This scale consists of 15 items and two sub-factors: emotional intimacy (9 items, e.g., My family feels intimate with each other.) and respect for acceptance (6 items, e.g., My family recognizes each other as they are.). Each item is rated on a five-point Likert scale ranging from 1 (not at all true) to 5 (very true). The internal consistency (Cronbach’s α) of the items was 0.97.

#### 2.2.4. Meaning in Life

Meaning in life was measured using a scale developed by Steger [93] and validated by Won, Kim, and Kwon [94] in the Korean version. This scale consists of 10 items and two sub-factors: existence of meaning (5 items, e.g., I know very well what makes my life meaningful.) and pursuit of meaning (5 items, e.g., I’m looking for the meaning in my life.). Each item is rated on a seven-point Likert scale ranging from 1 (not at all true) to 7 (very true). The internal consistency of the items (Cronbach’s α) was 0.93.

### 2.3. Procedure

Before conducting this study, the study was approved by the Institutional Review Board (protocol code JBNU 2022-06-026-001), and all research procedures were conducted ethically. Data were collected anonymously from all the participants.

### 2.4. Statistical Analysis

The data were analyzed using the IBM SPSS Statistics for Windows 26.0 and PROCESS Macro 3.5. The skewness and kurtosis of the data were checked using parametric statistical analysis. Pearson’s product–moment correlational analysis was conducted using SPSS and a sequential double moderating mediating effect was analyzed using PROCESS Macro 3.5 Model 6 [95]. Finally, bootstrapping using 5000 resamples with a 95% confidence interval was used to analyze the significance of mediating model.

## 3. Results

### 3.1. Correlation Analysis

Table 1 presents the correlational analysis of self-compassion, family support, family relationship satisfaction, and meaning in life of Korean baby boomers. None of the absolute values for skewness and kurtosis exceeded 2 and 7, respectively, indicating that all variables’ variances were close to the normal distribution for conducting parametric statistical analyses.

The correlational analysis revealed that self-compassion (*r* = 0.372, *p* < 0.01), meaning in life (*r* = 0.446, *p* < 0.01) and family support (*r* = 0.645, *p* < 0.01) were positively correlated with family relationship satisfaction of baby boomers. Family support was positively correlated with self-compassion (*r* = 0.421, *p* < 0.01) and meaning in life (*r* = 0.503, *p* < 0.01). Meaning in life was positively correlated with self-compassion (*r* = 0.524, *p* < 0.01).

### 3.2. Verification of the Double Mediation Model for the Meaning in Life of Baby Boomers

This study examined the mediating effects of self-compassion, family support, family relationship satisfaction, and meaning in life on Korean baby boomers (Table 2, Figure 2). It is known that statistical multicollinearity problems occur when tolerance is less than 0.2 or 0.1, and variance inflation factors (VIF) are greater than 5 or 10. Because the tolerance of predictors in this study were 0.545~0.805 and VIFs were 1.242~1.834, the multicollinearity problem was not significant. Additionally, the value of the Durbin–Watson statistic was 1.902, which indicates that there was no autocorrelation detected in the sample, as it was close to 2.

The results showed that self-compassion positively influenced family support (*B* = 0.638, *p* < 0.001), family support positively influenced family relationship satisfaction (*B* = 0.522, *p* < 0.001), and family relationship satisfaction positively influenced meaning in life (*B* = 0.329, *p* < 0.001). Through this, it is possible to confirm the mediating path of family support and family relationship satisfaction on self-compassion and the meaning in life.

Using 95% bootstrap confidence intervals from 5000 bootstrap replications, the double mediating effect of family support and family relationship satisfaction on the relationship between self-compassion and meaning in life of baby boomers was verified, and the results are presented in Table 3.

In this model, the total mediating effect was 0.330 (0.2190~0.4589), which was significant because there was no 0 between the upper and lower bounds of bootstrapping at 95% confidence intervals. Verifying the simple mediating effect revealed that the path from self-compassion to meaning in life via family support was significant (0.0143~0.2297). Furthermore, the path from self-compassion to meaning in life via family relationship satisfaction was also significant (0.0414~0.1973). Moreover, the sequential double mediating effect of family support and family relationship satisfaction on self-compassion and meaning in life (self-compassion → family support → family relationship satisfaction → meaning in life) was 0.110 (0.0490~0.2017), which was significant.

## 4. Discussion

### 4.1. Results and Implications

This study explored the relationships between self-compassion, family support, family relationship satisfaction, and meaning in life among Korean baby boomers. Further, it examined the double mediating effect of family support and family relationship satisfaction on self-compassion and meaning in life among baby boomers. These attempts have produced valuable information for further studies as well as for professionals who help baby boomers recover from suicidal ideation, and the implications are discussed below.

As a result of examining the relationship between self-compassion, family support, family relationship satisfaction, and the meaning in life of baby boomers through a correlation analysis, a significant correlation between all variables was found. Among them, self-compassion and family support showed a significant positive correlation, a finding in line with previous studies [60,61,62], which state that self-compassion has a positive correlation with support from others. These results can be interpreted as indicating that high self-compassion contributes to a positive attitude toward others as well as themselves, making them highly aware of familial support.

Family support and family relationship satisfaction showed a significant positive correlation, which is line with the findings of a previous study [84] that the higher the family support, the higher the satisfaction with family relationships. These results imply that family support, such as love, respect, and encouragement from families, which were considered only objects of care, are related to an increase in satisfaction with family relationships.

Second, significant mediating effects of family support on meaning in life of baby boomers was found. This means that self-compassion directly affects meaning in life and, simultaneously, affects meaning in life through family support. In other words, the higher the self-compassion of baby boomers, the higher their perception of family support, which increases their level of meaning in life. The continuous experience of self-compassion reduces self-criticism, allowing the maintenance of a healthy and calm state of mind [48,96]. This may lead to an improvement in self-regulation [97] and increase psychological capacity in the process of exploring and discovering meaning in life [98] to accept their past experiences without avoiding them.

In addition, because self-compassion involves treating oneself kindly with an open and uncritical attitude [42], it can be interpreted as having a positive effect on accepting support from others, that is, family members, beyond self-acceptance. Therefore, if baby boomers, who are increasingly criticizing themselves in old age [30], focus on positive self-confidence rather than just accepting family support, it can be expected to increase the level of meaning in life based on a balanced understanding of themselves.

Third, the mediating effect of family relationship satisfaction was significant in the relationship between self-compassion and the meaning in life of baby boomers. These results indicate that self-compassion directly affects meaning in life and simultaneously, affects meaning in life through family relationship satisfaction. In other words, the higher the self-compassion of baby boomers, the higher their satisfaction with family relationships and the higher their level of meaning in life. Baby boomers have the double burden of supporting parents and children economically. Consequently, it is also a generation that has failed to prepare for their retirement. Increasing family relationship satisfaction through interventions that increase self-compassion for them is likely to be a result of their efforts and hard work. It can be inferred that satisfaction increases the level of meaning in life.

Fourth, as a result of setting and analyzing the sequential double mediation model to see whether the self-compassion of baby boomers affects meaning in life through family support and family relationship satisfaction, both direct and indirect effects were significant. In other words, it was found that the higher the level of self-compassion of baby boomers, the higher the level of meaning in life, which supports previous studies [45,48]. These results suggest that mercy may be an important factor in enhancing meaning in life. In addition, an increase in the level of self-compassion increases family support, and high family support has a significant effect on raising the level of meaning in life, while positively evaluating past life by increasing family relationship satisfaction.

The results of this study indicate that in the relationship between self-compassion and meaning in life, not only regarding interventions to enhance family relationship satisfaction, but also family support is needed to escape from the loss of existence due to retirement. In other words, it was confirmed that the meaning in life cannot be increased simply by increasing family relationship satisfaction and that the level of meaning in life increases only through family support. These results show that the sequential mediating process of family support and family relationship satisfaction should precede the meaning in life.

In addition, the results of this study can be seen as supporting an intervention plan for reducing suicidal ideation described in the interpersonal psychological theory of suicide (IPTS) developed by Joiner [10]. It can be expected that suicidal ideation can be reduced by mediating the frustration of the desire to belong by raising the level of family support of the baby boomers and improving the level of self-compassion that positively evaluates their current life.

In the situation where mental health intervention measures are urgent due to the high suicide rate of the older adults in Korea, the development and implementation of a suicide prevention program for baby boomers that includes sessions to improve self-compassion, family support, and family relationship satisfaction level should be considered based on the implications of this study.

Older baby boomers are overwhelmed by negative evaluations due to a decrease in self-esteem, alienation, and regret, leading to self-criticism [29,30]. Currently, self-compassion intervention, which helps older adults understand that it is an aspect of life that they commonly experience, has a positive effect on emotional regulation and social bonds. In order to increase the level of self-compassion, training of the three sub-factors of self-compassion described by Neff [42] is required. The three sub-factors are as follows: (a) Self-kindness, which tolerates generously even if actions, emotions, and impulses feel inadequate. (b) Mindfulness to observe pain thoughtfully without inhibiting or exaggerating it. (c) Common humanity, in which negative experiences are not one’s own pain, but realize and accept that there is a similar vulnerability and pain in others.

If self-compassion intervention is repeated, openness to experience expands, and support from the family is perceived more. Since family support is provided in the form of visible types of behavior, help, services, and verbal and nonverbal information or advice, it is necessary to help baby boomers perceive and accept various forms of support from their families more clearly without misunderstanding [99]. In addition, since the relationship with adult children in the family relationship of old age has an important influence on the life satisfaction of the old age, a relationship promotion session for the relationship with adult children is required [100].

According to a systematic literature review study on suicide prevention programs for older adults in Korea [101], there are only two counseling programs for this demographic [102]. These programs focus on the education of transaction patterns, so there is a limit to applying them to the baby boomers. In addition, the currently used program was diagnosed as lacking in the theoretical framework, systematic composition, effectiveness verification, and the spread of the program during the development process. It has been suggested that it is important to develop and link an evidence-based program that can be applied in the actual field for the prevention of suicide in the older adults [101]. Therefore, considering the needs of the field and the proportion of baby boomers in old age, it is necessary to develop a program for suicide prevention among baby boomers based on the results of this study and to conduct follow-up studies so that they can be used in the field.

Efforts to promote re-employment may be considered to promote a sense of belonging and to encourage people to feel that they can still make productive contributions. Baby boomers are required to provide labor due to the nation’s lack of human resources in their youth, and there is a strong awareness that when they enter old age, they must continue to work due to insufficient preparation for old age [103]. Although the level of education and income of baby boomers is higher than that of the previous generation, baby boomers often wish to find a new job because of their large expenditures and debts [104]. When considering the meaning of work in Korea, the reason why they want re-employment may be related to the effort to find the worth of life and self, beyond the point of view of simply working as a reward for labor [105]. Therefore, it is necessary to expand education and employment projects for baby boomers to preserve their economic difficulties and support them to lead productive and active lives in their old age.

### 4.2. Limitations and Future Direction of Study

This study had the following limitations. First, because this study was conducted on 400 Korean baby boomers, the external validity could be limited.

Second, this study only examined the structural relationship between the self-compassion of the baby boomers and the variables set by the researcher, because few previous studies have been conducted. In the future, a follow-up study is needed to determine how self-compassion affects various psychological variables based on the basic data of this study.

Third, the effect of self-compassion on baby boomers was revealed, but it did not determine how to increase the level of self-compassion. Further research is needed to improve the level of self-compassion among these generations.

## 5. Conclusions

In Korea, it is urgent to come up with an intervention plan in terms of mental health due to the high suicide rate in old age. Considering the effect of depression on suicide, efforts are needed to increase the level of meaning in life that affects lowering depression. This is because the meaning in life contributes to improving psychological well-being by moving the attention of the elderly who focus on negative events experienced by aging to the psychological resources already acquired. Above all, since more than 700,000 baby boomers are entering the old age every year from 2020, it is important to take measures to raise the level of meaning in life for the baby boomers. In addition, the frustration regarding a sense of belonging and perceived burden on others are two important factors that make up the desire to commit suicide [10], so intervention in both areas is necessary. Baby boomers, who contributed more to productivity improvement than any other generation in the past, experience psychological threats, such as loss of existence as they face difficulties in productive contribution in old age. Additionally, as their social network shrinks, they feel alone. Therefore, the Korean government and researchers need to develop programs to reduce suicidal ideation among baby boomers. The results of this study suggest that educational programs that promote meaning in life are needed to reduce the suicidal ideation of the baby boomers. In addition, it is suggested that when developing a program, factors that increase the level of self-compassion and the level of family support and family relationship satisfaction should be included at the same time. This is because these factors inspire the feeling of belonging to family and reduce the perceived burden, thus the suicide rate can be lowered. On the other hand, efforts to activate re-employment opportunities for baby boomers are required. Because work allows us to experience the value of existence through a sense of belonging and productive contribution. These interventions are expected to help reduce the high suicide rate of Korean baby boomers and lead them to a healthier old age.

## Figures and Tables

**Figure 1 ijerph-19-09806-f001:**
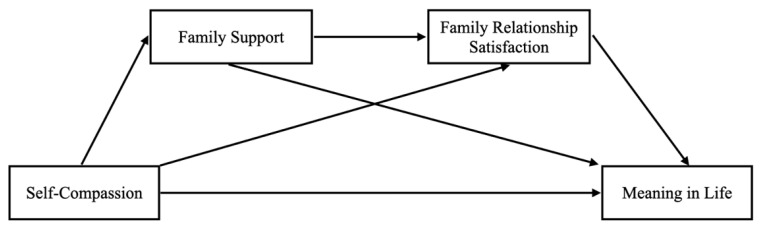
Double sequential mediation model.

**Figure 2 ijerph-19-09806-f002:**
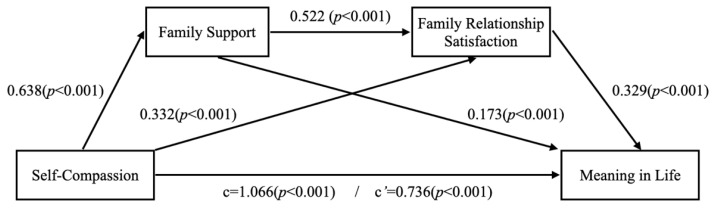
Examined double mediation model of family support and family relationship satisfaction on self-compassion and meaning in life of baby boomers.

**Table 1 ijerph-19-09806-t001:** Correlation coefficient of self-compassion, meaning in life, family support, and family relationship satisfaction (*N* = 400).

Variables	1	2	3	4
1. Self-compassion	-			
2. Meaning in life	0.524 **	-		
3. Family support	0.421 **	0.503 **	-	
4. Family relationship satisfaction	0.372 **	0.446 **	0.645 **	-
*M*	3.371	5.027	3.903	3.768
*SD*	0.503	1.024	0.863	0.795
Skewness	−0.060	−0.670	−0.822	−0.656
Kurtosis	0.375	0.777	0.242	0.548

** *p* < 0.01.

**Table 2 ijerph-19-09806-t002:** Double mediating effect of family support and family relationship on self-compassion and meaning in life of baby boomers.

Path	*B*	*S.E.*	*t*	LLCI	ULCI
Self-compassion → Family support	0.638	0.079	7.992 ***	0.481	0.795
Self-compassion → Family relationship satisfaction	0.332	0.063	5.268 ***	0.208	0.456
Family support → Family relationship satisfaction	0.522	0.036	14.207 ***	0.450	0.594
Self-compassion → Meaning in life	0.736	0.089	8.225 ***	0.560	0.912
Family support → Meaning in life	0.173	0.061	2.803 ***	0.051	0.295
Family relationship satisfaction → Meaning in life	0.329	0.068	4.785 ***	0.194	0.464

Note. *** *p* < 0.001. LLCI: lower level for confidence interval; ULCI: upper level for confidence interval.

**Table 3 ijerph-19-09806-t003:** Indirect effects of the mediation model.

Path	Effect	*S.E.*	BC 95% CI
Self-compassion → Family support → Meaning in life	0.110	0.055	0.0143~0.2297
Self-compassion → Family relationship satisfaction → Meaning in life	0.109	0.040	0.0414~0.1973
Self-compassion → Family support → Family relationship satisfaction → Meaning in life	0.110	0.039	0.0490~0.2017
Total indirect effect	0.330	0.061	0.2190~0.4589
Direct effect: Self-compassion → Meaning in life	0.736	0.089	0.5604~0.9124
Total effect	1.066	0.086	0.8957~1.2375

## Data Availability

Not applicable.

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
