# Peer review of "The Double Mediating Effect of Family Support and Family Relationship Satisfaction on Self-Compassion and Meaning in Life among Korean Baby Boomers"

_ijerph, 2022, doi:10.3390/ijerph19169806_

Round 1
Reviewer 1 Report
Please see attachment for details.

Author Response
"Please see the attachment."

Reviewer 2 Report
I may not be culturally qualified to review this article. It seems to me that Western audiences would not consider people born between 1955 and 1963 as elderly, particularly if they are still working and active, as most are, and certainly people of this age would not accept being labeled this way. Even though our culture is youth-obsessed, we make strong distinctions between the elderly as 75 + and those who are younger, and often consider even 75+ as not so old if they are still active.
That said, it is an interesting study and appears well-designed. There are some minor issues with Engish, and, again, some things may be somewhat lost in translation, more so culturally than linguistically.
The conclusion was a bit of a stretch, given we don't know how to improve self-compassion:
In a situation where mental health intervention measures are urgently needed due to 379 the high suicide rate of older adults in Korea, it is very important to raise the level of self- 380 compassion to raise the level of meaning in life, which affects the depression of baby boomers. Currently, in Korea, the proportion of older adults living alone is rapidly in-creasing, and the rate of lonely deaths due to psychological disconnection is also likely to increase. In this situation, if one inspires the feeling of belonging to the family through self-compassion and reduces the perceived burden, the suicide rate can be lowered. Hence, it is necessary for the Korean government to come up with an intervention plan 386 for the provision of psychological counseling services to improve the self-compassion 387 level of the baby boomers who are entering old age
Author Response
"Please see the attachment."

Reviewer 3 Report
I congratulate the authors for the clarity in the exposition of their manuscript. The presented study is well planned. The authors use good theoretical foundations and they present an interesting topic. I have some comments which might be helpful to improve it.
· Move the description of the sample with sociodemographic data to the "participants" section.
· In the description of the sample, the average age of the survey participants should be included.
· In the description of the measure, I suggest adding "sample items".
· The statistical analysis section should be supplemented with a description of correlation statistics and effect sizes taken.
· Were the relationships between the sociodemogrphic data and the variables analyzed in the study tested?
Author Response
"Please see the attachment."
